# The distribution and type B trichothecene chemotype of *Fusarium* species associated with head blight of wheat in South Africa during 2008 and 2009

**Gerhardus J. Van Coller**[1,2]*, **Lindy J. Rose**[2], **Anne-Laure Boutigny**[2], **Todd J. Ward**[3], **Sandra C. Lamprecht**[4], **Altus Viljoen**[2]

**1** Directorate: Plant Science, Western Cape Department of Agriculture, Elsenburg, South Africa,
**2** Department of Plant Pathology, Stellenbosch University, Matieland, South Africa, **3** United States Department of Agriculture–Agricultural Research Service, Peoria, Illinois, United States of America, **4** ARC-PHP, Stellenbosch, South Africa

* gert.vancoller@westerncape.gov.za

**Data Availability Statement:** All relevant data are within the manuscript and its Supporting information files.

## Abstract

Fusarium head blight (FHB) of wheat occurs commonly in irrigation regions of South Africa and less frequently in dryland regions. Previous surveys of *Fusarium* species causing FHB identified isolates using morphological characters only. This study reports on a comprehensive characterisation of FHB pathogens conducted in 2008 and 2009. Symptomatic wheat heads were collected from the Northern Cape, KwaZulu-Natal (KZN), Bushveld and eastern Free State (irrigation regions), and from one field in the Western Cape (dryland region). *Fusarium* isolates were identified with species-specific primers or analysis of partial *EF-1α* sequences. A representative subset of isolates was characterized morphologically. In total, 1047 *Fusarium* isolates were collected, comprising 24 species from seven broad species complexes. The *F. sambucinum* (FSAMSC) and *F. incarnatum-equiseti* species complexes (FIESC) were most common (83.5% and 13.3% of isolates, respectively). The *F. chlamydosporum* (FCSC), *F. fujikuroi* (FFSC), *F. oxysporum* (FOSC), *F. solani* (FSSC), and *F. tricinctum* species complexes (FTSC) were also observed. Within the FSAMSC, 90.7% of isolates belonged to the *F. graminearum* species complex (FGSC), accounting for 75.7% of isolates. The FGSC was the dominant *Fusaria* in all four irrigation regions. *F. pseudograminearum* dominated at the dryland field in the Western Cape. The Northern Cape had the highest species diversity (16 *Fusarium* species from all seven species complexes). The type B trichothecene chemotype of FGSC and related species was inferred with PCR. Chemotype diversity was limited (15-ADON = 90.1%) and highly structured in relation to species differences. These results expand the known species diversity associated with FHB in South Africa and include first reports of *F. acuminatum*, *F. armeniacum*, *F. avenaceum*, *F. temperatum*, and *F. pseudograminearum* from wheat heads in South Africa, and of *F. brachygibbosum*, *F. lunulosporum* and *F. transvaalense* from wheat globally. Potentially novel species were identified within the FCSC, FFSC, FOSC, FSAMSC, FIESC and FTSC.

**Funding:** This study was financially supported by the Western Cape Department of Agriculture (www.elsenburg.com), the Winter Cereal Trust (www.wintercerealtrust.co.za), the National Research Foundation (www.nrf.ac.za), and the United States Department of Agriculture - Agricultural Research Service National Program for Food Safety (www.usda.gov). The funders had no role in study design, data collection and analysis, decision to publish, or preparation of the manuscript.

**Competing interests:** The authors have declared that no competing interests exist.

## Introduction

Fusarium head blight (FHB) is a major disease of wheat (*Triticum aestivum*) worldwide. The disease reduces grain yield and causes the production of discoloured, shrivelled kernels contaminated with mycotoxins [1]. In the late 1990s, FHB resulted in losses estimated at US$ 2.7 billion in parts of the USA [2], while about 7 million ha have been affected by FHB epidemics in China [3]. The disease has also been damaging to wheat production in South America [4–7], Canada [8, 9] and Europe [10, 11]. In South Africa, wheat production has been negatively affected by the disease, although little information is available on its financial impact.

Studies conducted globally to identify the causal agents of FHB of wheat have demonstrated the *Fusarium graminearum* species complex (FGSC) to be widespread and predominant in many regions [9, 10, 12–14]. The FGSC, which is a subgroup within the *Fusarium sambucinum* species complex (FSAMSC) [15], consists of at least 16 phylogenetically distinct species [16, 17]. Members of the FGSC display significant biogeographic structure due to geographic speciation and host selection [12, 18–20]. Members of the FGSC can also infect barley (*Hordeum vulgare*), maize (*Zea mays*) and soybean (*Glycine max*); all crops that are grown in rotation with wheat in South Africa [21–23]. Other *Fusarium* species associated with FHB around the world include *F. chlamydosporum* [member of the *F. chlamydosporum* species complex (FCSC)]; *F. cerealis* (syn. *F. crookwellense*), *F. culmorum*, *F. poae*, and *F. pseudograminearum*, (members of the FSAMSC); *F. equiseti*, [member of the *F. incarnatum–equiseti* species complex (FIESC)]; *F. avenaceum* and *F. tricinctum* [members of the *F. tricinctum* species complex (FTSC)] and *F. proliferatum*, *F. subglutinans* and *F. verticillioides* [part of the *Fusarium fujikuroi* species complex (FFSC)] [10, 14, 15, 24–30].

*Fusarium* species associated with FHB produce mycotoxins, which are toxic secondary metabolites harmful to humans and animals. The most important among these include the type A and B trichothecene mycotoxins, and zearalenone (ZEA) [31]. Important type A trichothecene mycotoxins include diacetoxyscirpenol (DAS), neosolaniol (NEO), and T-2 and HT-2 toxins, while important type B trichothecene mycotoxins include deoxynivalenol (DON), and nivalenol (NIV) [31]. DON and ZEA (a nonsteroidal estrogen), are widely considered as the most important for wheat and barley [32], although NIV is also found in these crops [26]. The trichothecenes are potent inhibitors of eukaryotic protein synthesis and immunomodulatory [31] and are phytotoxic [33]. DON has two acetylated forms, namely 3-acetyldeoxynivalenol (3-ADON) and 15-acetyldeoxynivalenol (15-ADON) [32], while NIV has an acetylated form called fusarenon X (Fus-X) [10]. ZEA, on the other hand, has estrogenic properties associated with hyperestrogenism and infertility in pigs [31].

There are three major wheat production areas in South Africa [34]. These include the Western Cape Province (Mediterranean climate with mostly winter rainfall); the summer rainfall areas of Gauteng, Limpopo, Mpumalanga, and the North-West Province, the Northern Cape Province and KwaZulu-Natal (KZN); and the Free State Province (also summer rainfall). Wheat is also produced on small scale in the Eastern Cape Province. Historical reports of FHB in South Africa emanated from the irrigated production regions of the eastern Free State, North West Province and KZN during the 1980s [35], with only one report from dryland fields in the Western Cape [36]. FHB of wheat in KZN and the North West Province was shown to be caused by *Gibberella zeae* (now FGSC), and in the eastern Free State by *F. crookwellense* (*F. cerealis*) [35]. The Northern Cape was considered disease-free. This changed a few years later when FHB reached epidemic proportions in irrigated wheat fields near the Orange River in the Northern Cape during the early 1990s [37]. At one locality, 28% of the grain samples were infected with *Fusarium* species, including *F. graminearum* (member of FGSC) (48.4%), *F. moniliforme* (36.3%) (now *F. verticillioides*) and *F. subglutinans* (1.6%) (members of FFSC), *F.*

*equiseti* (member of FIESC) (9.7%), *F. chlamydosporum* (member of FCSC) (3.2%), and *F. oxysporum* (member of *F. oxysporum* species complex, FOSC) (0.8%). *Fusarium poae* was reported from glume spot of wheat heads in South Africa in 1996, in association with a mite species (*Siteroptes avenae*) [38]. A later study [19] identified a total of 277 *Fusarium* isolates, designated FGSC in the current study, and found that *F. graminearum s.s.* with the 15-ADON chemotype was the dominant member associated with FHB in South Africa. The largest species diversity occurred in KZN, where five of the six FGSC members in South Africa was found. A study published in 2017 [39] reported *F. graminearum* (member of FGSC) as the most common *Fusarium* spp. causing FHB of wheat in seven localities across four South African provinces, although no mention is made of when this study was conducted. Only five other *Fusarium* species were identified, including *F. chlamydosporum* (member of FCSC) and *F. equiseti* (member of FIESC) in four localities in the Northern Cape, and *F. cerealis* and *F. culmorum* (members of FSAMSC), and *F. semitectum* (member of FIESC) at one locality each in the Eastern Cape Province, KZN and North West Province. The only records of FHB from dryland fields in the Western Cape Province was on diseased wheat heads under dryland conditions from three farms in the George-district and one farm in the Swellendam-district, where the causal organism was identified as *F. graminearum* Group 2 (now FGSC) [36].

*Fusarium* spp. of wheat can differ between regions and fields due to a combination of climatic factors; agronomical practices such as crop rotation, tillage practices and the amount and type of stubble; host genotype; and disease management practices, which include host resistance and chemical control [10, 40–44]. Consequently, the *Fusarium* species occurring on wheat in a country or region can change in response to these drivers. For instance, *F. avenaceum*, *F. culmorum* and *F. poae* were historically common in the colder regions of northern Europe, but the FGSC has become more dominant in some of these regions in recent years, believed to be due to an increase in maize production and climate change [11, 25, 45]. In Italy, on the other hand, *F. graminearum* is replaced by *F. poae*, believed to be due to variation in environmental conditions [46]. Furthermore, the homothallic nature of the FGSC allows for the mass production of ascospores, which may aid in the epidemiology of the disease [47].

The commercial release of new spring wheat cultivars able to complete its life cycle in a shorter period than older cultivars enabled producers in the irrigation regions to sequentially cultivate crops like wheat in the winter and maize in the summer on the same fields, a practice known as double-cropping [48], which is done by producers in the irrigation regions to this day. Irrigation practices like flood-irrigation have also been replaced with centre-pivot irrigation, which creates a more suitable microclimate for FHB development [49]. Conservation agriculture, which includes minimum / no-tillage practices and crop rotation with barley, oats and broad-leaf crops like canola replaced conventional tillage and monoculture in the Western Cape [34, 50] from the late 1990s onwards. All of these practices may have impacted the occurrence of FHB and associated *Fusarium* species in South Africa. With the availability of powerful molecular techniques, numerous new *Fusarium* species and species complexes were described [51]. These techniques facilitated reassessments of species identity among isolates from major reference collections [52] and enabled accurate identification of fusaria via curated databases of DNA sequence data [53]. With the exception of one study, which focused only on the FGSC [19], all previous studies of *Fusarium* species associated with FHB of wheat in South Africa described isolates using morphological characteristics only. Previous surveys, furthermore, collected diseased wheat heads from only a limited number of localities within large and geographically diverse production regions. As such, there is a need to have both an updated and phylogenetically broader understanding of FHB pathogen diversity within South Africa. The aim of this study, therefore, was to conduct a survey to determine the identity, distribution and type B trichothecene chemotype of *Fusarium* species associated with FHB of wheat in

South Africa. This survey was conducted during 2008 and 2009, and to the best of our knowledge, no subsequent surveys has since been undertaken or published from South Africa. The findings of this study, therefore, provides valuable new information and also serves as a foundation for new studies to be conducted in future.

## Materials and methods

### Wheat production areas in South Africa

Spring wheat cultivars are grown under dryland conditions in the Western Cape Province and under centre-pivot irrigation in the summer rainfall irrigation regions and Free State Province. Winter wheat is grown under dryland (rain-fed) conditions in the Free State Province (Fig 1). The Western Cape Province is the largest production area in the country, although it yields less grain ha$^{-1}$ than the irrigation areas. There are two production regions in the Western Cape, namely the Swartland (western) and the Overberg (southern) regions. Regions within the summer rainfall irrigation area differ greatly in terms of climate and soil type. It consists of the Bushveld; which comprises parts of Gauteng, Limpopo, Mpumalanga and the North West

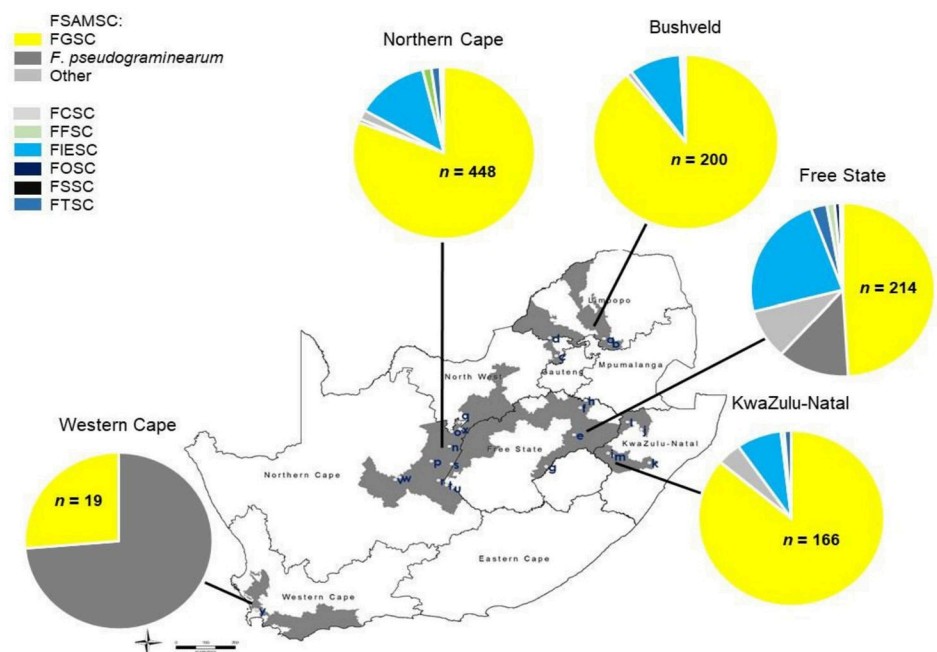

**Fig 1. Geographical distribution of *Fusarium* species obtained from diseased wheat heads in South Africa during 2008 and 2009, according to the total number of *Fusarium* isolates obtained (*n*).** Bushveld: a = Marble Hall, b = Groblersdal, c = Brits, d = Koedoeskop; Free State: e = Bethlehem, f = Frankfort, g = Ladybrand, h = Villiers; KwaZulu-Natal, i = Bergville, j = Dundee, k = Greytown, l = Newcastle, m = Winterton; Northern Cape: n = Barkly West, o = Bull Hill, p = Douglas, q = Hartswater, r = Hopetown, s = Modderrivier, t = Orania 1, u = Orania 2, v = Prieska, w = Remhoogte, x = Vaalharts; Western Cape: y = Vissershok. FSAMSC = *F. sambucinum* species complex. FGSC = *Fusarium graminearum* species complex, species observed *include F. graminearum*, *F. boothii*, *F. meridionale*, *F. acaciae-mearnsii*, *F. brasilicum*, and *F. cortaderiae*. Other FSAMSC = members of the FSAMSC other than FGSC and *F. pseudograminearum*: *F. armeniacum*, *F. brachygibbosum*, *F. cerealis*, *F. culmorum*, *F. lunulosporum*, *F. poae*, and unknown *Fusarium* species within the FSAMSC. FCSC = *Fusarium chlamydosporum* species complex: *F. chlamydosporum* clade 1 and clade 5 (O'Donnell *et al.*, 2009), and unknown *Fusarium* species within the FCSC. FFSC = *Fusarium fujikuroi* species complex: *F. subglutinans*, *F. temperatum*, *F. verticillioides*, and unknown *Fusarium* species within the FFSC. FIESC = *Fusarium incarnatum-equiseti* species complex. FOSC = *Fusarium oxysporum* species complex: *Fusarium oxysporum*, and unknown *Fusarium* species within the FOSC. FSSC = *F. solani* species complex: *F. solani* clade 5 (O'Donnell *et al.*, 2009; Zhang *et al.*, 2006). FTSC = *Fusarium tricinctum* species complex: *F. acuminatum*, *F. avenaceum*, and unknown *Fusarium* species within the FTSC.

Province; the eastern part of the Free State Province (referred to hereafter as the Free State); parts of KZN; and areas in the Northern Cape Province in the vicinity of the Orange-, Vaal- and Modder River (referred to hereafter as the Northern Cape).

## Collection of wheat heads

**Irrigation production areas.** Wheat heads with FHB symptoms were collected at 15 localities in the Free State, KZN and the Northern Cape during 2008, and at 14 localities in the Bushveld, Free State, KZN, and Northern Cape during 2009 (Fig 1; Table 1). The collection sites did not overlap between years, and although localities like Bethlehem and Ladybrand (Free State), Dundee and Winterton (KZN), and Remhoogte (Northern Cape) did overlap between years, wheat heads were sampled from different fields at these localities over the two years. Collections were done in the irrigation spring wheat cultivar evaluation trials of the

**Table 1. Production region, cultivar, crop history and geographical information of localities where wheat heads with Fusarium head blight symptoms were sampled.**

| Sampling year | Production region | Locality | Previous crop[a] | GPS coordinates | Elevation (m) |
|---|---|---|---|---|---|
| 2008 | Free State | Bethlehem | Fallow | 28.161474˚ S 28.304801˚ E | 1643 |
| | | Frankfort | Maize | 27.178925˚ S 28.405266˚ E | 1489 |
| | | Ladybrand | Cabbage | 29.184615˚ S 27.553087˚ E | 1536 |
| | | Villiers | Maize | 27.038234˚ S 28.662245˚ E | 1528 |
| | KwaZulu-Natal | Bergville | Soybean | 28.753716˚ S 29.342924˚ E | 1127 |
| | | Winterton | Maize | 28.885823˚ S 29.471612˚ E | 1119 |
| | | Dundee | Maize | 28.104323˚ S 30.262652˚ E | 1214 |
| | Northern Cape | Douglas | Groundnut | 29.012244˚ S 23.960668˚ E | 1006 |
| | | Hartswater | Maize | 27.795774˚ S 24.779891˚ E | 1105 |
| | | Modderrivier | Maize | 29.104098º S 24.579221º E | 1128 |
| | | Prieska | Maize | 29.609747º S 22.856582º E | 938 |
| | | Orania 1 | Maize | 29.790727º S 24.423845º E | 1098 |
| | | Orania 2 | Maize | 29.881946˚ S 24.585153˚ E | 1123 |
| | | Remhoogte | Maize | 29.537154º S 22.993977º E | 980 |
| | | Vaalharts | Maize | 27.965495˚ S 24.836811˚ E | 1176 |
| 2009 | Bushveld | Brits | Sunflower | 25.593013˚ S 27.768504˚ E | 1087 |
| | | Groblersdal | Peppers | 25.178062˚ S 29.389781˚ E | 936 |
| | | Koedoeskop | Soybean | 25.011394˚ S 27.562786˚ E | 955 |
| | | Marble Hall | Cabbage | 25.041904˚ S 29.221597˚ E | 927 |
| | Free State | Bethlehem | Soybean | 28.161434˚ S 28.305021˚ E | 1643 |
| | | Ladybrand | Cabbage | 29.181017˚ S 27.556070˚ E | 1545 |
| | KwaZulu-Natal | Dundee | Maize | 27.984337˚ S 30.349992˚ E | 1168 |
| | | Greytown | Fallow | 29.084172˚ S 30.603934˚ E | 1028 |
| | | Newcastle | Soybean | 27.643130˚ S 29.979236˚ E | 1192 |
| | | Winterton | Soybean | 28.839872˚ S 29.467234˚ E | 1097 |
| | Northern Cape | Barkly West | Onions | 28.507932˚ S 24.593006˚ E | 1109 |
| | | Bull Hill | Maize | 28.048725˚ S 24.579655˚ E | 1060 |
| | | Hopetown | Maize | 29.636905º S 24.176112º E | 1071 |
| | | Remhoogte | Maize | 29.538249˚ S 22.969875˚ E | 961 |
| | Western Cape | Vissershok | Canola[b] | 33.785813˚ S 18.555610˚ E | 12 |

[a] Crop grown during the previous summer growing season, except where indicated

[b] Crop grown during the previous winter growing season (summer fallow)

Agricultural Research Council's Small Grain (ARC-SG). A total of 20 samples were collected from each of four cultivars (Baviaans, Duzi, Kariega and PAN3434), which were planted in a randomised block design with four replicates, to provide 80 samples per locality. Additionally, 40 symptomatic wheat heads were sampled randomly from one locality each in the Free State in 2008 (Frankfort), and KZN in 2009 (Greytown). The former was from a commercial wheat field, where collections were made from the spring cultivar SST835, while the latter was from a field demonstration trial from a local seed company, where collections were made from the spring cultivar PAN3434.

**Western Cape Province.** The dryland spring wheat cultivar evaluation trials of the ARC-SG were inspected each year at three localities each in the Overberg and Swartland production regions of the Western Cape Province (dryland production) for the presence of FHB, but no visible disease was found. Visible FHB symptoms was, however, found in a commercial dryland wheat field (Vissershok) during 2009, where 40 symptomatic wheat heads were randomly sampled from the spring cultivar SST027.

## Isolations from diseased kernels

For the first year, two visually scabby kernels per sample were surface-disinfected by washing in 70% ethanol for 1 min, followed by 1 min in a 1% sodium hypochlorite solution. The kernels were then rinsed with sterile distilled water and left to air dry in the laminar flow cabinet on sterile tissue paper. One kernel was plated onto potato dextrose agar (PDA) (Biolab Diagnostics, Midrand, South Africa) amended with 40 mg $L^{-1}$ streptomycin sulphate, and the other kernel onto selective *Fusarium* agar (SFA) [54]. During the second year, only one visually scabby kernel per sample was surface-disinfected as described above, and isolated onto PDA only.

Plates were incubated for 5 days at 21˚C in the dark. Developing *Fusarium* colonies were purified and single-spored. Single-spore cultures were plated onto PDA to harvest mycelium for DNA extraction, and onto divided plates containing PDA amended with 40 mg $L^{-1}$ streptomycin sulphate and carnation leaf agar (CLA) [54], which was incubated underneath cool white and near-UV lights with a photoperiod of 12 hrs for 21 days, for morphological identification [54]. Single-spore cultures were stored in 15% glycerol at -80˚C at the Department of Plant Pathology, Stellenbosch University, South Africa.

## Fungal reference cultures

Reference isolates of *F. graminearum* (NRRL28439) and *F. culmorum* (NRRL3288) were obtained from Dr K. O'Donnell (USDA-ARS Peoria, IL, USA), while isolates of *F. avenaceum* (MRC 3227) and *F. poae* (MRC 8486) were provided by Prof W.F.O Marasas (MRC-PRO-MEC, Tygerberg, South Africa). The reference isolates for *F. cerealis* (MRC 8399; CAV359) and FIESC (MRC 1813; CAV367) were provided by Prof A. Viljoen (Department of Plant Pathology, Stellenbosch University, South Africa), while the positive control for *F. pseudograminearum* (WCA3532) was obtained in this study following identification with multilocus genotyping (MLGT) by Dr T.J. Ward (USDA-ARS Peoria, IL, USA).

Isolates with known type B trichothecene chemotype identities were provided by Laëtitia Pinson-Gadais (French National Institute for Agricultural Research, Villenave d'Ornon, France). These included *F. culmorum* with a 3-ADON chemotype (INRA 233), FGSC with a 15-ADON chemotype (INRA 156) and FGSC with a NIV chemotype (INRA 91). These isolates served as positive controls in PCR assays to determine the chemotype of B-trichothecene isolates obtained in this study.

## Identification of *Fusarium* isolates

Single-spore cultures were grown on PDA plates for 7 days, where after genomic DNA was extracted from mycelia using the Wizard® SV Genomic DNA Purification System Kit (Promega, South Africa). Isolates of *F. avenaceum*, *F. culmorum*, FGSC and *F. poae* were identified in a multiplex PCR with known species-specific primers (Table 2). Amplifications were carried out with an initial denaturation step at 94˚C for 2 min, followed by 35 cycles of denaturation at 94˚C for 45 s, annealing at 60˚C for 30 s and extension at 72˚C for 45 s, with a final extension step of 72˚C for 5 min [22]. Isolates of *F. cerealis* and *F. pseudograminearum* were also identified by PCR with species-specific primers listed in Table 2, using the same reaction conditions [22]. Isolates that did not generate PCR products were identified by sequencing of the translation elongation factor-1 alpha (*EF-1α*) gene [55]. These included members of the FGSC. The *EF-1α* gene was amplified in a PCR reaction that consisted of an initial denaturation step of 94˚C for 5 min, followed by 30 cycles of denaturation at 94˚C for 30 s, primer annealing at 57˚C for 45 s and primer extension at 72˚C for 1 min, followed by a final extension of 72˚C for 7 min [22]. Generated *EF-1α* products were purified and sequenced, and edited sequences were compared to sequences available on the NCBI GenBank database (https://www.ncbi.nlm.nih.gov/genbank/), the CBS-KNAW Fungal Biodiversity Centre's *Fusarium* MLST website (http://www.cbs.knaw.nl/Fusarium), and the FUSARIUM-ID database [56]. Isolates with less than 99% sequence similarity to reference sequences in *Fusarium* MLST were annotated as unknown species (F. sp.) and may represent novel species-level diversity. If the same *Fusarium* species was obtained from the two kernels of the same sample on the different growing media, only one isolate was selected to represent the sample. DNA sequence data generated for 45

**Table 2. Primer names, sequences and expected sizes of PCR products of *Fusarium* species associated with head blight of wheat.**

| Target | Primer | Primer sequence (5'–3') | Annealing T (˚C) | Amplicon size (bp) | Reference |
|---|---|---|---|---|---|
| *F. avenaceum* | FaF | CAAGCATTGTCGCCACTCTC | 60 | 920 | [57] |
| | FaR | GTTTGGCTCTACCGGGACTG | | | |
| *F. cerealis* | CroAF | CTCAGTGTCCACCGCGTTGCGTAG | 60 | 842 | [58] |
| | CroAR | CTCAGTGTCCCAATCAAATAGTCC | | | |
| *F. culmorum* | Fc01F | ATGGTGAACTCGTCGTGGC | 60 | 570 | [59] |
| | Fc01R | CCCTTCTTACGCCAATCTCG | | | |
| FGSC[a] | Fg11F | CTCCGGATATGTTGCGTCAA | 60 | 450 | [59] |
| | Fg11R | GGTAGGTATCCGACATGGCAA | | | |
| FIESC[b] | FeF1 | CATACCTATACGTTGCCTCG | 60 | 400 | [60] |
| | FeR1 | TTACCAGTAACGAGGTGTATG | | | |
| *F. poae* | Fp82F | CAAGCAAACAGGCTCTTCACC | 60 | 220 | [61] |
| | Fp82R | TGTTCCACCTCAGTGACAGGTT | | | |
| *F. pseudograminearum* | FpgF | GTCGCCGTCACTATC | 60 | 779 | [62] |
| | FpgR | CACTTTATCTCTGGTTGCAG | | | |
| *EF1α* | EF1 | ATGGGTAAGGA(A/G)GACAAGAC | 57 | 648 | [55] |
| | EF2 | GGA(G/A)GTACCAGT(G/C)ATCATGTT | | | |
| *Tri3* | 3CON | TGGCAAAGACTGGTTCAC | 58 | 243 (3-ADON) | [63] |
| | 3NA | GTGCACAGAATATACGAGC | | 610 (15-ADON) | [63] |
| | 3D15A | ACTGACCCAAGCTGCCATC | | 840 (NIV) | [63] |
| | 3D3A | CGCATTGGCTAACACATG | | | [63] |

[a] *Fusarium graminearum* species complex

[b] *Fusarium incarnatum–equiseti* species complex

isolates from FHB in South Africa have been deposited in GenBank under accession numbers MG588054–MG588069, MG588071–MG588087, MK617767–MK617769, and MK629641–MK629649. The identification of FIESC isolates by *EF-1α* gene sequencing were confirmed via PCR with species-specific primers listed in Table 2. Amplifications were carried out with an initial denaturation step at 95˚C for 2 min, followed by 35 cycles of denaturation at 95˚C for 45 s, annealing at 60˚C for 30 s and extension at 72˚C for 1 min, with a final extension step of 72˚C for 2 min [22].

The identities of a representative group of type B trichothecene-producing isolates were confirmed at the USDA-ARS (Peoria, IL, USA) using a multilocus genotyping assay (MLGT) [63]. These included *F. cerealis* (five isolates), *F. culmorum* (one isolate), *F. pseudograminearum* (two isolates), *Fusarium lunulosporum* (three isolates), while 277 isolates of the FGSC obtained in this study were previously identified [19]. Of the 277 FGSC isolates, 85.2% were identified as *F. graminearum s.s.* An additional 32 FGSC isolates were also identified at the USDA-ARS (Peoria, IL, USA) as *F. boothii*, *F. graminearum s.s.* and *F. meridionale* using the MLGT assay. In the current study, FGSC isolates were only identified using the FGSC-specific primer-pair mentioned earlier [59], and the FGSC species will therefore not be referred to by their phylogenetic species names. The molecular identities of FHB isolates from South Africa were linked to prior morphological species definitions by studying the morphology of 85 isolates representing all fusaria obtained [54]. The identity of six isolates could not be determined morphologically. These included three isolates of *F. transvaalense*, one isolate of *F. brachygibbosum*, and one isolate of an unknown *Fusarium* species (FSAMSC), as well as one isolate of *F. temperatum* (FFSC).

## Chemotype identification

The NIV, 3-ADON and 15-ADON chemotypes of FGSC and related species within clade 1 of the FSAMSC (FSAMSC-10) [64] were identified using a multiplex PCR that amplified portions of the *Tri3* gene [63] (Table 2). The PCR reaction conditions consisted of an initial denaturation step at 94˚C for 2 min, followed by 35 cycles of denaturation at 94˚C for 30 s, primer-annealing at 58˚C for 30 s and extension at 72˚C for 30 s, with a final extension step of 72˚C for 5 min.

## Results

### *Fusarium* species

A total of 1047 *Fusarium* isolates were identified in this study (S1 Table), which included 24 *Fusarium* species from seven major *Fusarium* species complexes (Table 3). The FSAMSC accounted for 83.5% of all isolates, with most FSAMSC isolates belonging to the FGSC subgroup. The FGSC comprised 69.1% (*n* = 439) of *Fusarium* isolates obtained in 2008 and 85.9% (*n* = 354) in 2009. The FIESC accounted for 13.3% of all isolates, and the other species complexes (FFSC, FTSC, FOSC, FCSC, and FSSC) each accounted for less than 1.5% of the FHB isolates. Due to the use of clade-specific primers in this study, 487 of the 874 FSAMSC isolates were identified only to the level of the FGSC. However, 14 named species were identified among the remaining 387 FSAMSC isolates. Among the FCSC, we identified two informally named species (FCSC 1 and FCSC 5). Among the FFSC, we identified *F. subglutinans*, *F. temperatum*, and *F. verticillioides*. *F. oxysporum* and the informally named species FSSC 5 were identified from the FOSC and FSSC respectively. *F. acuminatum* and *F. avenaceum* were identified from the FTSC. All but two of the 139 FIESC isolates were identified with clade-specific primers that did not permit species level identification. The two FIESC isolates identified via DNA sequence analyses, as well as 11 additional isolates from the other species complexes

**Table 3. Incidence[a, b] of *Fusarium* isolates obtained from diseased wheat heads in South Africa during 2008 and 2009.**

| Locality / region | FCSC[c] | | | FFSC[d] | | | | FIESC[e] | FOSC[f] | | FSAMSC[g] | | | | | | | | | | FSSC[h] | FTSC[i] | | | Total[j] |
|---|---|---|---|---|---|---|---|---|---|---|---|---|---|---|---|---|---|---|---|---|---|---|---|---|---|
| | FCSC 1 | FCSC 5 | F. sp. | sub | temp | vert | F. sp. | | oxy | F. sp. | arm | bra | cer | cul | FGSC | lun | poae | pse | tvl | F. sp. | FSSC 5 | acu | ave | F. sp. | |
| **2008** | | | | | | | | | | | | | | | | | | | | | | | | | |
| *Free State (FS)* | | | | | | | | | | | | | | | | | | | | | | | | | |
| Bethlehem | 0 | 0 | 0 | -[k] | 0 | 0 | - | 27 | 0 | 1 | 0 | - | 23 | 3 | 0 | 0 | 0 | 39 | 0 | - | 0 | 0 | 3 | 0 | 51 |
| Frankfort | 0 | 0 | 0 | - | 0 | 0 | - | 0 | 0 | 0 | 0 | - | 8 | 0 | 91 | 0 | 0 | 0 | 0 | - | 0 | 0 | 0 | 0 | 24 |
| Ladybrand | 0 | 0 | 0 | - | 0 | 2 | - | 66 | 0 | 2 | 0 | - | 4 | 2 | 13 | 0 | 0 | 0 | 0 | - | 0 | 0 | 8 | 0 | 45 |
| Villiers | 0 | 0 | 0 | - | 2 | 0 | - | 6 | 0 | 0 | 0 | - | 0 | 0 | 90 | 0 | 0 | 0 | 0 | - | 1 | 0 | 0 | 0 | 80 |
| **Total FS** | **0** | **0** | **0** | **-** | **1** | **0.5** | **-** | **24** | **0** | **1** | **0** | **-** | **8** | **1** | **50** | **0** | **0** | **10** | **0** | **-** | **0.5** | **0** | **3** | **0** | **200** |
| *KwaZulu-Natal (KZN)* | | | | | | | | | | | | | | | | | | | | | | | | | |
| Bergville | 0 | 0 | 0 | - | 0 | 0 | - | 5 | 0 | 0 | 0 | - | 0 | 0 | 86 | 0 | 2 | 0 | 0 | - | 0 | 2 | 2 | 0 | 36 |
| Dundee | 0 | 0 | 0 | - | 0 | 0 | - | 9 | 0 | 0 | 0 | - | 0 | 0 | 90 | 0 | 0 | 0 | 0 | - | 0 | 0 | 0 | 0 | 32 |
| Winterton | 0 | 0 | 0 | - | 0 | 0 | - | 36 | 0 | 0 | 0 | - | 0 | 0 | 63 | 0 | 0 | 0 | 0 | - | 0 | 0 | 0 | 0 | 11 |
| **Total KZN** | **0** | **0** | **0** | **-** | **0** | **0** | **-** | **11** | **0** | **0** | **0** | **-** | **0** | **0** | **84** | **0** | **1** | **0** | **0** | **-** | **0** | **1** | **1** | **0** | **79** |
| *Northern Cape (NC)* | | | | | | | | | | | | | | | | | | | | | | | | | |
| Douglas | 0 | 0 | 0 | - | 0 | 0 | - | 30 | 0 | 0 | 0 | - | 0 | 0 | 57 | 0 | 0 | 7 | 0 | - | 2 | 0 | 0 | 2 | 40 |
| Hartswater | 0 | 0 | 0 | - | 0 | 0 | - | 12 | 12 | 0 | 0 | - | 0 | 0 | 62 | 0 | 0 | 0 | 0 | - | 0 | 0 | 12 | 0 | 8 |
| Modderrivier | 0 | 20 | 6 | - | 0 | 0 | - | 53 | 0 | 0 | 6 | - | 0 | 0 | 0 | 0 | 0 | 0 | 6 | - | 0 | 6 | 0 | 0 | 15 |
| Orania 1 | 1 | 0 | 0 | - | 1 | 0 | - | 5 | 0 | 0 | 0 | - | 0 | 0 | 88 | 0 | 0 | 0 | 1 | - | 0 | 0 | 1 | 1 | 75 |
| Orania 2 | 0 | 0 | 0 | - | 0 | 0 | - | 3 | 0 | 0 | 0 | - | 0 | 0 | 94 | 0 | 0 | 0 | 1 | - | 0 | 0 | 0 | 0 | 55 |
| Prieska | 0 | 0 | 0 | - | 0 | 0 | - | 17 | 0 | 0 | 0 | - | 0 | 0 | 75 | 5 | 0 | 0 | 0 | - | 0 | 0 | 0 | 1 | 52 |
| Remhoogte | 0 | 0 | 0 | - | 0 | 0 | - | 1 | 0 | 0 | 0 | - | 0 | 0 | 98 | 0 | 0 | 0 | 0 | - | 0 | 0 | 0 | 0 | 59 |
| Vaalharts | 0 | 0 | 0 | - | 0 | 0 | - | 38 | 0 | 0 | 0 | - | 0 | 0 | 55 | 0 | 0 | 0 | 0 | - | 0 | 0 | 1 | 0 | 52 |
| **Total NC** | **0.3** | **3** | **0.3** | **-** | **0.3** | **0** | **-** | **16** | **0.3** | **0** | **0** | **-** | **0** | **0** | **76** | **0.8** | **0** | **0.8** | **0.8** | **-** | **0.3** | **0.3** | **0.8** | **0.8** | **356** |
| **Total 2008[j]** | **1** | **5** | **1** | **-** | **3** | **1** | **-** | **115** | **1** | **2** | **1** | **-** | **16** | **3** | **439** | **3** | **1** | **23** | **3** | **-** | **2** | **2** | **10** | **3** | **635** |
| **2009** | | | | | | | | | | | | | | | | | | | | | | | | | |
| *Bushveld* | | | | | | | | | | | | | | | | | | | | | | | | | |
| Brits | - | - | - | 2 | - | - | 0 | 38 | - | 0 | - | 5 | 0 | - | 52 | - | 0 | 0 | - | 0 | - | - | - | - | 34 |
| Groblersdal | - | - | - | 0 | - | - | 0 | 1 | - | 0 | - | 0 | 0 | - | 98 | - | 0 | 0 | - | 0 | - | - | - | - | 64 |
| Koedoeskop | - | - | - | 0 | - | - | 0 | 7 | - | 0 | - | 0 | 0 | - | 92 | - | 0 | 0 | - | 0 | - | - | - | - | 53 |
| Marble Hall | - | - | - | 0 | - | - | 0 | 0 | - | 2 | - | 0 | 0 | - | 97 | - | 0 | 0 | - | 0 | - | - | - | - | 49 |
| **Total Bushveld** | **-** | **-** | **-** | **0.5** | **-** | **-** | **0** | **9** | **-** | **0.5** | **-** | **1** | **0** | **-** | **89** | **-** | **0** | **0** | **-** | **0** | **-** | **-** | **-** | **-** | **200** |
| *Free State* | | | | | | | | | | | | | | | | | | | | | | | | | |
| Bethlehem | - | - | - | 0 | - | - | 0 | 0 | - | 0 | - | 0 | 11 | - | 11 | - | 0 | 77 | - | 0 | - | - | - | - | 9 |
| Ladybrand | - | - | - | 0 | - | - | 0 | 20 | - | 0 | - | 0 | 0 | - | 80 | - | 0 | 0 | - | 0 | - | - | - | - | 5 |
| **Total FS** | **-** | **-** | **-** | **0** | **-** | **-** | **0** | **7** | **-** | **0** | **-** | **0** | **7** | **-** | **35** | **-** | **0** | **50** | **-** | **0** | **-** | **-** | **-** | **-** | **14** |
| *KZN* | | | | | | | | | | | | | | | | | | | | | | | | | |
| Dundee | - | - | - | 0 | - | - | 0 | 0 | - | 0 | - | 0 | 0 | - | 100 | - | 0 | 0 | - | 0 | - | - | - | - | 29 |
| Greytown | - | - | - | 0 | - | - | 4 | 13 | - | 0 | - | 0 | 0 | - | 63 | - | 4 | 0 | - | 13 | - | - | - | - | 22 |
| Newcastle | - | - | - | 0 | - | - | 0 | 5 | - | 0 | - | 0 | 0 | - | 85 | - | 10 | 0 | - | 0 | - | - | - | - | 20 |

*(Continued)*

**Table 3.** (Continued)

| Locality / region | FCSC[c] | | | FFSC[d] | | | | FIESC[e] | FOSC[f] | | FSAMSC[g] | | | | | | | | | | FSSC[h] | FTSC[i] | | | Total[j] |
|---|---|---|---|---|---|---|---|---|---|---|---|---|---|---|---|---|---|---|---|---|---|---|---|---|---|
| | FCSC 1 | FCSC 5 | F. sp. | sub | temp | vert | F. sp. | | oxy | F. sp. | arm | bra | cer | cul | FGSC | lun | poae | pse | tvl | F. sp. | FSSC 5 | acu | ave | F. sp. | |
| Winterton | - | - | - | 0 | - | - | 0 | 0 | - | 0 | - | 0 | 0 | - | 100 | - | 0 | 0 | - | 0 | - | - | - | - | 16 |
| **Total KZN** | | | | **0** | | | **1** | **4** | | **0** | | **0** | **0** | | **87** | | **3** | **0** | | **3** | | | | | **87** |
| *Northern Cape* | | | | | | | | | | | | | | | | | | | | | | | | | |
| Barkly West | - | - | - | 0 | - | - | 0 | 0 | - | 0 | - | 0 | 0 | - | 100 | - | 0 | 0 | - | 0 | - | - | - | - | 52 |
| Bull Hill | - | - | - | 0 | - | - | 0 | 0 | - | 0 | - | 0 | 3 | - | 96 | - | 0 | 0 | - | 0 | - | - | - | - | 33 |
| Hopetown | - | - | - | 0 | - | - | 0 | 20 | - | 0 | - | 0 | 0 | - | 80 | - | 0 | 0 | - | 0 | - | - | - | - | 5 |
| Remhoogte | - | - | - | 0 | - | - | 0 | 0 | - | 0 | - | 0 | 0 | - | 100 | - | 0 | 0 | - | 0 | - | - | - | - | 2 |
| **Total NC** | | | | **0** | | | **0** | **1** | | **0** | | **0** | **1** | | **97** | | **0** | **0** | | **0** | | | | | **92** |
| *Western Cape (WC)* | | | | | | | | | | | | | | | | | | | | | | | | | |
| Vissershok | - | - | - | 0 | - | - | 0 | 0 | - | 0 | - | 0 | 0 | - | 26 | - | 0 | 73 | - | 0 | - | - | - | - | 19 |
| **Total WC** | | | | **0** | | | **0** | **0** | | **0** | | **0** | **0** | | **26** | | **0** | **73** | | **0** | | | | | **19** |
| **Total 2009[j]** | | | | **1** | | | **1** | **24** | | **1** | | **2** | **2** | | **354** | | **3** | **21** | | **3** | | | | | **412** |

[a] Incidence for a locality was calculated as follows: (number of isolates of a *Fusarium* species obtained at a locality / number of *Fusarium* isolates obtained at the locality) x 100

[b] Incidence for a production region was calculated as follows: (number of isolates of a *Fusarium* species obtained in a production region / number of *Fusarium* isolates obtained in the production region) x 100

[c] FCSC = *Fusarium chlamydosporum* species complex: FCSC 1, FCSC 5 = *F. chlamydosporum* clade 1 and clade 5 (O'Donnell *et al*., 2009); F. sp. = unknown *Fusarium* species within the FCSC

[d] FFSC = *Fusarium fujikuroi* species complex: sub = *F. subglutinans*; temp = *F. temperatum*; vert = *F. verticillioides*; F. sp. = unknown *Fusarium* species within the FFSC

[e] FIESC = *Fusarium incarnatum-equiseti* species complex

[f] FOSC = *Fusarium oxysporum* species complex: oxy = *Fusarium oxysporum*; F. sp. = unknown *Fusarium* species within the FOSC

[g] FSAMSC = *Fusarium sambucinum* species complex: arm = *F. armeniacum*; bra = *F. brachygibbosum*; cer = *F. cerealis*; cul = *F. culmorum*; FGSC = *Fusarium graminearum* species complex, species observed include *F. graminearum, F. boothii, F. acaciae-mearnsii,F. brasilicum*, and *F. cortaderiae*; lun = *F. lunulosporum*; poae = *F. poae*; pse = *F. pseudograminearum*; tvl = *F. transvaalense*; F. sp. = unknown *Fusarium* species within the FSAMSC

[h] FSSC = *F. solani* species complex: FSSC 5 = *F. solani* clade 5 (O'Donnell *et al*., 2009; Zhang *et al*., 2006)

[i] FTSC = *Fusarium tricinctum* species complex: acu = *F. acuminatum*; ave = *F. avenaceum*; F. sp. = unknown *Fusarium* species within the FTSC

[j] Total number of *Fusarium* isolates

[k] *Fusarium* species not obtained in specific year

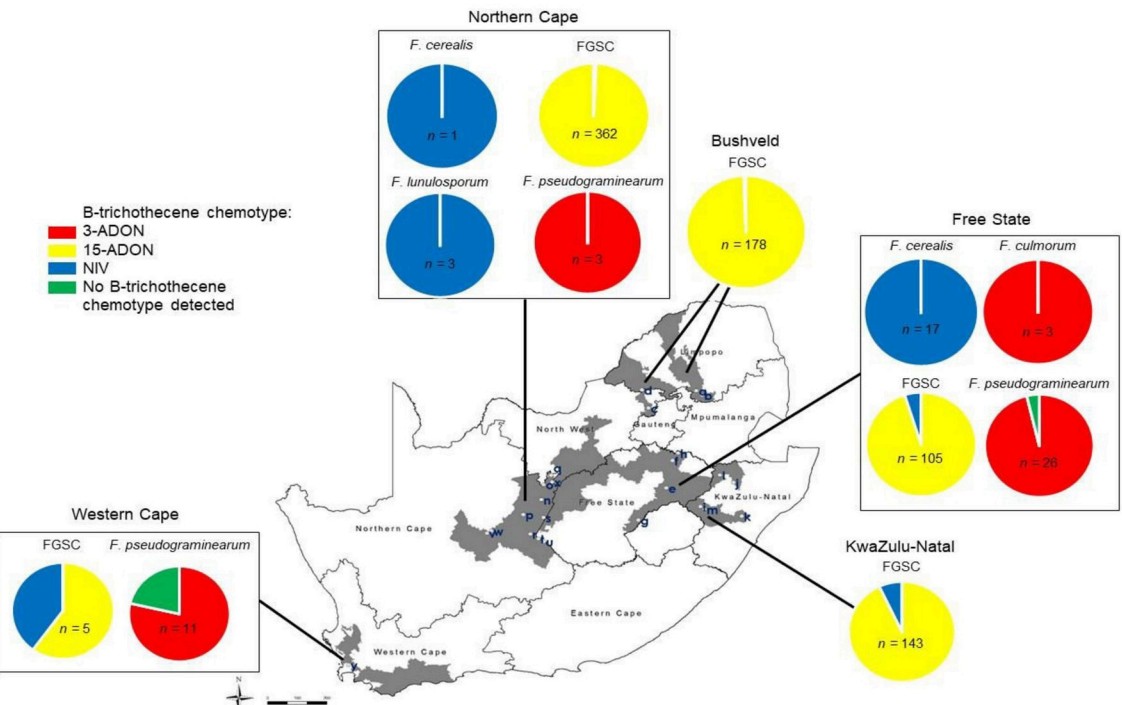

**Fig 2. Incidence of type B trichothecene chemotypes of different members of the *Fusarium sambucinum* species complex (FSAMSC) obtained from diseased wheat heads in South Africa during 2008 and 2009, according to the total number of FSAMSC isolates obtained (*n*).** Bushveld: a = Marble Hall, b = Groblersdal, c = Brits, d = Koedoeskop; Free State: e = Bethlehem, f = Frankfort, g = Ladybrand, h = Villiers; KwaZulu-Natal; i = Bergville, j = Dundee, k = Greytown, l = Newcastle, m = Winterton; Northern Cape: n = Barkly West, o = Bull Hill, p = Douglas, q = Hartswater, r = Hopetown, s = Modderrivier, t = Orania 1, u = Orania 2, v = Prieska, w = Remhoogte, x = Vaalharts; Western Cape: y = Vissershok. FGSC = *Fusarium graminearum* species complex, species observed include *F. graminearum*, *F. boothii*, *F. meridionale*, *F. acaciae-mearnsii*, *F. brasilicum*, and *F. cortaderiae*. B-trichothecene chemotype: 3-ADON = 3-acetyldeoxynivalenol, 15-ADON = 15-acetyldeoxynivalenol, NIV = Nivalenol, no B-trichothecene chemotype detected = *F. pseudograminearum* isolates that did not produced a result with PCR to indicate its chemotype.

lacked similarity to reference sequences in the Fusarium MLST database sufficient for species identification (F. sp) and may represent novel species diversity (Table 3).

Considerably more *Fusarium* isolates were obtained in 2008 than in 2009, even though the same number of diseased wheat heads were collected in both years. This can be attributed to the high incidence of *Alternaria* isolates found in wheat kernels at localities like Brits (Bushveld), Remhoogte, Bull Hill and Hopetown (Northern Cape), KZN, the Free State and the Western Cape in 2009 (data not presented). Many symptomatic kernels from these regions yielded no *Fusarium* isolates during 2009. All members of the FCSC and FTSC, *F. temperatum*, *F. verticillioides*, *F. oxysporum*, *F. armeniacum*, *F. culmorum*, *F. lunulosporum*, *F. transvaalense*, and FSSC 5 were collected only in 2008. *F. subglutinans*, the *F.* sp. isolate from the FFSC, *F. brachygibbosum*, and the *F.* sp. isolates from the FSAMSC were collected only in 2009 (Table 3).

Co-occurrence of *Fusarium* species (when more than one *Fusarium* species was obtained from the same wheat head or same kernel), was very low and occurred only in 0.9% of all isolations performed. Of these cases, the FIESC co-occurred 48.4% (*n* = 16) with the FGSC, and 16.1% (*n* = 5) with other fusaria.

## Geographical distribution of *Fusarium* species

The FGSC and FIESC were the most widely distributed fusaria in South Africa (Fig 1). The FGSC was found in all production regions and localities, apart from Modderrivier (Northern Cape) and Bethlehem (Free State) in 2008 (Table 3). *Fusarium pseudograminearum* was predominant at the one locality in the Western Cape. The incidence of the FGSC was highest at Remhoogte (98.3%) and Orania 2 (94.6%) in the Northern Cape, at Frankfort (91.7%) and Villiers (90%) in the Free State, and at Dundee (90.6%) in KZN in 2008. In 2009, its incidence was highest at Dundee and Winterton in KZN, and Barkly West in the Northern Cape, all at 100%, followed by Groblersdal (98.4%) and Marble Hall (98%) in the Bushveld. Ladybrand and Bethlehem in the Free State had the lowest incidence of the FGSC in 2008 and 2009, at 13.3% and 11.1% respectively (Table 3). In 2009, Remhoogte had an incidence of 100%, compared to 98.3% in 2008, although a total of 58 *Fusarium* isolates were obtained there in 2008 compared to just two isolates in 2009 (S1 Table).

The FIESC was obtained in all wheat production regions except the Western Cape. The fungus was not isolated at Frankfort (Free State) in 2008, and at several localities in 2009, including Bethlehem (Free State), Marble Hall (Bushveld), Dundee and Winterton (KZN), and Barkly West, Bull Hill and Remhoogte (Northern Cape) (Table 3). In the Free State, KZN, and the Northern Cape, FIESC comprised 24.5, 11.4 and 16% of isolates collected in 2008, respectively (Table 3). In 2009, the FIESC comprised 9, 7.1, 4.6 and 1.1% of isolates collected in the Bushveld, Free State, KZN and the Northern Cape, respectively. The highest incidence in 2008 was obtained at Ladybrand in the Free State (66.7%), followed by Modderrivier (53.3%) and Vaalharts (38.5%) in the Northern Cape (Table 3). The incidence of the FIESC was substantially reduced in 2009, with the highest incidence found at Brits in the Bushveld (38.2%). Where present, the lowest incidence of the FIESC in 2008 was at Remhoogte in the Northern Cape (1.7%), and at Groblersdal in the Bushveld (1.6%) in 2009 (Table 3).

The Northern Cape was the wheat production region with the highest FHB species diversity, with 16 *Fusarium* species from all seven species complexes obtained there (Fig 1). Ten *Fusarium* species from six species complexes were collected from wheat in the Free State, and seven *Fusarium* species from four species complexes in KZN. Five *Fusarium* species from four species complexes were obtained from wheat in the Bushveld, while only two species from one species complex (FSAMSC) were obtained at the locality in the Western Cape (Fig 1).

*Fusarium* species other than those in the FGSC and FIESC were mostly obtained at low incidences (Table 3). The members of the FCSC, *F. oxysporum* within the FOSC, *F. armeniacum*, *F. lunulosporum*, *F. transvaalense* and an unknown *Fusarium* species within the FTSC were obtained only in the Northern Cape in 2008. *Fusarium temperatum* and FSSC 5 were obtained only in the Free State and Northern Cape in 2008, while *F. verticillioides* and *F. culmorum* were obtained only in the Free State in 2008. *Fusarium acuminatum* and *F. avenaceum* was obtained from KZN and the Northern Cape, and from the Free State, KZN and Northern Cape in 2008 respectively. *Fusarium subglutinans* and *F. brachygibbosum* were only obtained in 2009 in the Bushveld, while unknown *Fusarium* spp. within the FFSC and FSAMSC were obtained only in 2009 in KZN. An unknown *Fusarium* sp. within the FOSC was obtained in the Free State in 2008, and from the Bushveld in 2009. *Fusarium cerealis* was obtained from the Free State in 2008 and 2009, and from the Northern Cape in 2009 only, while *F. poae* was only obtained in KZN, where it occurred both years. *Fusarium pseudograminearum* was obtained both years in the Free State, and in the Northern and Western Cape during 2009 (Table 3).

The greatest species diversity at individual localities in 2008 was found at Ladybrand (Free State) and Orania 1 (Northern Cape) with seven species each, and the lowest at Dundee and

Winterton (KZN), Frankfort (Free State), and Remhoogte (Northern Cape), with two species each. In 2009, the highest species diversity was at Greytown (KZN) with five species, and the lowest at Dundee and Winterton (KZN), and Barkly West and Remhoogte (Northern Cape), where only the FGSC was obtained (Table 3).

## Type B trichothecene chemotype

The B-trichothecene chemotypes of 861 isolates from FSAMSC-1 were assessed via a chemotype-specific PCR assay. 15-ADON was the dominant type (90.1%) associated with these *Fusarium* isolates collected in South Africa (S1 Table). Isolates with the 3-ADON and NIV types comprised 5.4 and 4.5%, respectively, of the FSAMSC-1 isolates in the country. The 15-ADON type was only observed among the FGSC, where it was predominant (97.4%). Less than 0.5% of this species complex had the 3-ADON type, while 2.3% had the NIV type (S1 Table). *Fusarium cerealis* and *F. lunulosporum* were exclusively of the NIV type, and *F. culmorum* and *F. pseudograminearum* were exclusively of the 3-ADON type (Fig 2).

## Geographic distribution of type B trichothecene chemotypes

*Fusarium* species representing all three B-trichothecene chemotypes were present in all wheat production regions of South Africa, except for KZN and the Bushveld, where the 3-ADON type was absent (Fig 2). Four *Fusarium* species or species complexes with B-trichothecene chemotypes were collected in the Northern Cape (*F. cerealis*, FGSC, *F. lunulosporum*, *F. pseudograminearum*) and Free State (*F. cerealis*, *F. culmorum*, FGSC and *F. pseudograminearum*) (Fig 2). FGSC was the sole fusaria with a B-trichothecene chemotype among the FSAMSC-1 isolates in the Bushveld and KZN, while both the FGSC and *F. pseudograminearum* in the Western Cape had B-trichothecene chemotypes (Fig 2).

15-ADON was most dominant type found in all production regions, mainly due to the widespread occurrence of the FGSC in South Africa. Of the FGSC isolates obtained in the Northern Cape and Bushveld, more than 99% were of the 15-ADON type, while more than 95% of FGSC isolates obtained in the Free State and more than 90% of FGSC isolates obtained in KZN were of the 15-ADON type. Since *F. pseudograminearum* dominated at Vissershok in the Western Cape, the 3-ADON type was dominant there (78.6%) (Fig 2).

## Discussion

Twenty-four *Fusarium* species from seven of the major *Fusarium* species complexes were associated with FHB of wheat in South Africa during 2008 and 2009. Species from the FGSC (part of FSAMSC) were most common. This confirms previous reports on the dominance of FGSC as FHB pathogens in South Africa [35–37, 39] and internationally [1, 9–11, 13, 14, 26, 45]. In the 1980s, only the FGSC was obtained from diseased wheat heads in South Africa in KZN and parts of the Bushveld, while *F. cerealis* was found in the eastern parts of the Free State [35]. Seed batches from FHB-infected wheat fields at Prieska (Northern Cape) collected 10 years later provided the first reports of *F. verticillioides* (formerly *F. moniliforme*) and *F. subglutinans*, *F. equiseti*, *F. chlamydosporum* and *F. oxysporum* associated with FHB in South Africa [37]. *F. poae* was reported from glume spot of wheat heads in South Africa in 1996 [38]. *F. culmorum* and *F. semitectum* was added to the list of *Fusarium* species associated with wheat heads in a more recent report [39].

Six *Fusarium* species are reported here for the first time to be associated with FHB of wheat in South Africa. These include *F. acuminatum*, *F. armeniacum*, *F. avenaceum*, *F. temperatum*, *F. poae* and *F. pseudograminearum*. Some *Fusarium* species from wheat heads were also reported for the first time in certain production regions. *Fusarium cerealis* was found for the

first time in the Northern Cape; *F. culmorum* in the Free State; the FIESC in the Bushveld, Free State and KZN; and *F. oxysporum* (FOSC) in the Northern Cape. Although this is the first report of FCSC 1 and FCSC 5 [28], and FSSC 5 [65, 66] from wheat grain globally, the species complexes to which they belong (FCSC and FSSC) have been reported from wheat previously, including in South Africa [10, 37]. Based on sequencing data of the *EF1-α* gene-area, unknown *Fusarium* species were also obtained from FCSC, FFSC, FOSC, FSAMSC, and FTSC. The identity of these species will be determined in subsequent studies.

All members of FCSC, FFSC, FOSC, FSSC and FTSC, as well as *F. armeniacum*, *F. brachygibbosum*, *F. culmorum*, *F. lunulosporum*, *F. poae*, *F. transvaalense* (FSAMSC) and an unknown species within the FSAMSC were obtained at low frequencies in this study, which indicate them to be of minor importance as FHB pathogens in South Africa. All these species, apart from *F. brachygibbosum*, *F. lunulosporum* and *F. transvaalense* have previously been associated with wheat globally [10, 24–26, 29, 67]. *Fusarium brachygibbosum* has been reported to cause stalk rot of maize in China [68] and has been obtained from diseased human tissue [28]. *Fusarium lunulosporum* was first isolated from grapefruit exported to Europe from South Africa in 1968, and the species was formally described in 1977 [69]. Although this species has a type B trichothecene (NIV) chemotype, its infrequent occurrence on wheat in South Africa makes it potentially a less important FHB-pathogen. *Fusarium transvaalense* was recently described from rhizosphere soil in the Kruger National Park in South Africa [70], and the present study is, to our knowledge, the first report of this species from wheat globally.

At the time of this survey, *F. cerealis* has been replaced as the primary pathogen of wheat in the eastern Free State [35] by the FGSC in the north and *F. pseudograminearum* and the FIESC in the south. This may have been due to an increase in maize production and warmer temperatures in the area, which has been shown to favour the FGSC over cold-weather pathogens such as *F. culmorum* [11, 25, 45]. The FGSC was also shown to be a more effective DON producer than the closely related *F. culmorum* [71], and is homothallic, which may aid in the epidemiology of the pathogen [47]. Double-cropping of wheat and maize may have introduced the FGSC into the region. In this study, *F. cerealis* was still found in the eastern Free State, and more frequently than in any other part of South Africa. During this time period, the eastern Free State also yielded the highest levels of the FIESC, which was partly due to the high incidence of the species complex at Ladybrand, especially during 2008, when 66.7% of isolates obtained at this location belonged to the FIESC. The dominance of this species complex at Ladybrand may be ascribed to the cropping history at the time, which consisted of wheat rotated with cabbage, since mulch of cruciferous crops like white mustard (*Sinapis alba*) and Indian mustard (*Brassica juncea*) have been reported to suppress *Fusarium* infection and decrease mycotoxin contents in wheat grain [72]. The dominance of FIESC at this locality may, therefore, have been due to the relative absence of the FGSC, brought about by the crop rotation practice, since this was the only locality where wheat was rotated with a cruciferous crop. The FIESC did, however, occur at several other localities, at frequencies varying from 1.56–53.33%. The FIESC was first reported from FHB in South Africa in grain samples obtained from FHB infected wheat fields near Prieska in the Northern Cape [37]. The FIESC was also the Fusaria co-occurring most frequently with other species, although co-occurrence of Fusaria in the same wheat head or kernel was very low (0.9% of isolations performed). The reason for the relative high occurrence of the FIESC during 2008 and 2009 is unclear, but may be due to sampling conducted at the dough stage (Zadoks growth stage 83–85), when FHB symptoms are most visible, but kernels are not fully developed. When performing isolations, it can be unclear which kernels are diseased when they are dry a few days after sampling. A subsequent study on the FIESC isolates obtained in this study revealed high species diversity, but

low toxigenic potential (unpublished data), indicating that this species complex may be less important as FHB pathogens in South Africa.

An interesting observation was the dominance of *F. pseudograminearum* as an FHB pathogen at one locality in the eastern Free State (both years) and the one locality in the Western Cape (2009). The dominance of this species at Bethlehem in the eastern Free State may be due to it being introduced earlier via infected seed [35, 85], especially since *F. pseudograminearum* was not found at any other locality in this region. Although the sample size in the Western Cape was quite small (one locality with 40 wheat heads sampled), a study conducted more recently from three localities in the Western Cape revealed that *F. pseudograminearum* was the dominant species obtained from wheat heads exhibiting FHB symptoms at all three localities, constituting more than 80% of ~300 isolates obtained (unpublished data). *Fusarium pseudograminearum* is best known as the cause of Fusarium crown rot (FCR) of wheat [73, 74]. Its dominance in the Western Cape can be ascribed to the prevalence of FCR in this region as well as the use of minimum / no till practices, which results in a build-up of inoculum levels in stubble [74]. The *F. pseudograminearum* isolates obtained in this study belong exclusively to the 3-ADON chemotype, which reflects results from Australia [27], Canada [75] and China [14]. It is, however, unclear whether the difference in chemotype may be the reason why *F. pseudograminearum* dominated over the FGSC (15-ADON) at Bethlehem. The superior ability of *F. pseudograminearum* to cause FCR has been ascribed to its ability to produce higher levels of DON than *F. culmorum* and *F. graminearum* in the stem base, while *F. culmorum* and *F. graminearum* produced high levels of DON in grains to cause FHB [76]. However, an outbreak of FHB in Australia was shown to be caused by both *F. graminearum* and *F. pseudograminearum*, indicating a lack of specialisation for FHB among these species [27]. Since the epidemiology of FHB and FCR differs drastically [8, 73], the question arises whether some level of specialisation is not present in the genetically highly diverse *F. pseudograminearum* population [77].

The vast majority of FGSC isolates in this study had the 15-ADON chemotype, with a few exceptions. This corresponds to results obtained from barley in the Northern Cape of South Africa [22], and from wheat in Argentina [5], Brazil [78], parts of Europe, and China [25, 45]. FGSC isolates with the NIV chemotype dominated at Greytown in KZN, while 40% of the FGSC isolates collected at Vissershok in the Western Cape had the NIV chemotype. FGSC isolates with the 3-ADON chemotype were found in three adjacent localities in the Northern Cape. It is important that *Fusarium* mycotoxins and their acetylated forms be determined during surveillance studies, as these might provide insights on the distribution of toxigenic forms of the fungus. Between 1999 and 2000, a small, localised populations of the FGSC with the 3-ADON chemotype was discovered in the Midwestern-USA, which might have been introduced to this region [79]. A 14-times increase in *F. graminearum s.s.* with the 3-ADON chemotype was subsequently reported in western Canada between 1998 and 2004 [63]. Strains from this introduced population produced significantly more DON and had a higher growth rate and fecundity than the population characterised by the 15-ADON type, therefore posing a significant threat to food safety and security. This difference in toxin accumulation and aggressiveness between *F. graminearum s.s.* isolates with the 3-ADON vs 15-ADON chemotype is, however, likely related to differences in the genetics of the two populations, and not a direct result of trichothecene chemotype differences [63, 80].

Crop rotation and tillage practices can partly account for the differences in *Fusarium* species composition and diversity within production regions and localities in this study. Double cropping of maize and wheat is standard practice under conventional tillage in most summer rainfall wheat production regions, while wheat and maize / soybean are frequently produced under no-till conditions in KZN [81, 82]. Minimum / no-till practices, which include

minimum soil disturbance, crop rotation and soil coverage with stubble or living plants [50], is also common in the Western Cape [34]. Although minimum / no-till practices hold various advantages for producers and the environment, it does result in an increase in the amount of stubble left on the soil, which can subsequently increase the risk of stubble-borne diseases like FHB and FCR [42, 44, 83]. In a study on the colonization of residues of different plant species by *F. graminearum* and their contribution to Fusarium head blight inoculum in Uruguay [84], it was found that the FGSC was more frequently isolated from residues of wheat and barley than residues of sunflower or *Festuca arundinacea* (tall fescue). The FGSC produced more ascospores, the primary source of inoculum for FHB, in wheat and barley residues than maize or other gramineous hosts, while not producing any on sunflower residues. The FGSC further-more survived longer on wheat and barley residues under no-tillage production compared to reduced tillage production. Finally, some level of specialisation in the association between *Fusarium* species and type of stubble was found. *Fusarium avenaceum* and *F. sambucinum*, for example, was isolated from wheat, barley and gramineous stubble, but not from sunflower or tall fescue. In this study, FGSC was sometimes found to be the dominant species at localities where the previous crop was not maize, and where conventional tillage was practiced. These included all localities in the Bushveld, where the FGSC was abundant and the previous crops at the respective sites were sunflower, peppers, soybean and cabbage. A comprehensive study to determine the incidence and severity of FHB in different crop rotation systems and tillage regimes is, therefore, recommended. This, along with the use of host resistance and chemical control, can form part of an integrated disease management approach.

The absence of FHB in the Northern Cape during the 1980s [35] can partly be attributed to the practice of wheat production followed by a fallow-period in the summer, coupled with removal of stubble and conventional tillage, in addition to flood irrigation. The introduction of FHB of wheat to the Northern Cape is unknown. The replacement of old with new wheat cultivars from 1988 to 2008 [34], coupled with the introduction of double-cropping, could have introduced the disease with infected seed [35, 85]. A population genetics study of the most important members of the FGSC population in all production regions of South Africa, as was done in a study of *F. graminearum* isolates from Canada and the USA [86] may elucidate the origin of the disease in the Northern Cape region and the rest of South Africa.

FGSC and FIESC isolates were more abundant in the Free State in 2008 compared to 2009. This also happened for the FGSC in the Northern Cape. In KZN, the occurrence of the FGSC remained almost unchanged during the two years. The presence of all members of FCSC, *F. temperatum* and *F. verticillioides*, *F. oxysporum* (FOSC), *F. armeniacum*, *F. culmorum*, *F. lunu-losporum*, *F. transvaalense*, FSSC 5, *F. acuminatum*, *F. avenaceum*, and an unknown *Fusarium* sp. within FTSC in 2008 but not 2009, and *F. subglutinans*, *F. brachygibbosum*, and unknown *Fusarium* spp. within FFSC and FSAMSC in 2009 but not 2008, may have been coincidental since the incidence of all of these species was very low. The high incidence of *Alternaria* species obtained in all production regions in 2009 might have contributed to lower *Fusarium* levels in the 2009 production season, while differences in climate, cropping history and agronomic practices of the collection sites in the two years might have also contributed to the discrepancy in *Fusarium* species composition between years [40, 41, 43]. Variation in the timing and the amount of water provided through irrigation, especially near anthesis, could also have influ-enced the resultant disease intensity and associated *Fusarium* species between years [87]. Reli-able disease forecasting models to aid producers in managing the disease [1], therefore, need to be developed for South African wheat producers.

Using MLGT, the identity and type B trichothecene chemotype of 277 FGSC isolates obtained in this study was determined [19]. This study, however, extends and places into con-text the previous results by reporting on the identity of all the Fusaria associated with FHB of

wheat in South Africa obtained in 2008 and 2009. It showed that FHB pathogens of wheat were wide-spread in South Africa, and that the diversity of *Fusarium* species associated with FHB was greater than previously reported [19, 35–39]. Given the time since this study was conducted, changes in the *Fusarium* species associated with FHB and distribution of these species may have occurred in response to, among others, changes in crop production practices, environment, and the level of resistance of cultivars. Future surveys are needed to ascertain which *Fusarium* species are currently dominant.

This study also reported on the type B trichothecene chemotype profile (3-ADON, 15-ADON and NIV) of *F. cerealis*, *F. culmorum*, FGSC, *F. lunulosporum* and *F. pseudograminearum* associated with FHB of wheat in South Africa. FGSC was the dominant contributor to FHB and contained the only isolates with the 15-ADON type, the most prevalent trichothecene type observed. The dominance of the FGSC at almost every locality sampled in South Africa indicates that the local grain industry is at risk of contamination of grain with well-known mycotoxins such as DON, NIV and ZEA [10]. Legislation on Maximum Tolerated Levels of DON was introduced in South Africa in 2016 [88]. More research is thus needed to determine the amount of DON and ZEA in harvested grain over different seasons and at different localities across South Africa. This could be achieved by quantifying fungal biomass of type B trichothecene producing *Fusarium* species under natural conditions in South Africa using real-time quantitative PCR, and by correlating this with mycotoxin levels in harvested grain [23]. The type B trichothecene mycotoxins and ZEA are, however, not the only important *Fusarium* mycotoxins occurring in harvested grain in South Africa. Follow-up studies must therefore also be conducted to determine the level of contamination of harvested grain with other mycotoxins like the type A trichothecenes DAS and NEO [31], as well as mycotoxins produced by *Alternaria* species [89].

## Supporting information

**S1 Table. Strain data for 1047 Fusarium isolates obtained from wheat plants with FHB symptoms in different production areas in South Africa.**
(XLSX)

## Acknowledgments

We thank E. Nowers and Z. Sedeman (Western Cape Department of Agriculture), and N. Orwig and T. Usgaard (United States Department of Agriculture—Agricultural Research Service National Program for Food Safety) for excellent technical support; and the Western Cape Department of Agriculture, the Department of Plant Pathology at Stellenbosch University, and the United States Department of Agriculture—Agricultural Research Service National Program for Food Safety for supporting infrastructure. Mention of trade names or commercial products in this article is solely for the purpose of providing specific information and does not imply recommendation or endorsement by either the Western Cape Department of Agriculture, Stellenbosch University, or the United States Department of Agriculture (USDA). The Western Cape Department of Agriculture, Stellenbosch University and the USDA are equal opportunity providers and employers.

## Author Contributions

**Conceptualization:** Gerhardus J. Van Coller, Altus Viljoen.

**Data curation:** Gerhardus J. Van Coller.

**Formal analysis:** Gerhardus J. Van Coller.

**Funding acquisition:** Gerhardus J. Van Coller, Altus Viljoen.

**Investigation:** Gerhardus J. Van Coller, Todd J. Ward, Sandra C. Lamprecht, Altus Viljoen.

**Methodology:** Gerhardus J. Van Coller, Lindy J. Rose, Anne-Laure Boutigny, Todd J. Ward, Sandra C. Lamprecht, Altus Viljoen.

**Project administration:** Gerhardus J. Van Coller, Altus Viljoen.

**Resources:** Gerhardus J. Van Coller, Lindy J. Rose, Anne-Laure Boutigny, Todd J. Ward, Sandra C. Lamprecht, Altus Viljoen.

**Software:** Lindy J. Rose, Anne-Laure Boutigny, Todd J. Ward.

**Supervision:** Altus Viljoen.

**Validation:** Altus Viljoen.

**Visualization:** Gerhardus J. Van Coller.

**Writing – original draft:** Gerhardus J. Van Coller.

**Writing – review & editing:** Lindy J. Rose, Anne-Laure Boutigny, Todd J. Ward, Sandra C. Lamprecht, Altus Viljoen.

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
