## [Decision Letter · Decision Letter 0]

16 Nov 2021

PONE-D-21-28634The distribution and type B trichothecene chemotype of *Fusarium* species associated with head blight of wheat in South AfricaPLOS ONE

Dear Authors

Thank you for submitting your manuscript to PLOS ONE. After careful consideration, we feel that it has merit but does not fully meet PLOS ONE’s publication criteria as it currently stands. Therefore, we invite you to submit a revised version of the manuscript that addresses the points raised during the review process.

Please address the comments raised by both reviewers.

We look forward to receiving your revised manuscript.

Kind regards,

Kris Audenaert, Ph.D.

Academic Editor

PLOS ONE

Journal Requirements:

2. In your Methods section, please provide additional location information about your study sites, including geographic coordinates for the data set if available

Additional Editor Comments (if provided):

Dear authors,

Based on the comments of one reviewer and my own review, I suggest to amend the paper with minor revisions.

Both the reviewer and myself appreciate the dimenssion of the sampling campaign.

some minor comments from my side:

1. Please (if necessary) change the significant numbers in table 3, when less than 100 samples are taken, you cannot have a % incidence of lower than 1%...so something like 36.36 is impossible, should be then 36

2. The chemotypes presented in a table is not clear. It would be better to somehow also include a pie diagram put on the geography map. It is known from previous work by Waalwijk and coworkers that geography (e.g. altitutde) can affect the chemotype.

3. There are quite some differences between the western cape , free state, and the rest...the reason for these differences are now somehow addressed in the discussion, but in my opinion would merit a section in the results.

4. DId you look at co-occurence? recent insights (study by tan et al., 2021 and some reports before by several authors) report on the tendency of some species to co-occur in disease complexes. Did you observe any of these co-occurences as they might be one of the important drivers explaining how the plasticity of a population works.

5. What about climatic conditions? can your report be put in a historic context? have you seen changes compared to previous report?

Please commend on these remarks and on the remarks raised by the second reviewer.

kind regards

Reviewers' comments:

Reviewer's Responses to Questions

**Comments to the Author**

1. Is the manuscript technically sound, and do the data support the conclusions?

Reviewer #1: Yes

2. Has the statistical analysis been performed appropriately and rigorously? 

Reviewer #1: Yes

3. Have the authors made all data underlying the findings in their manuscript fully available?

Reviewer #1: Yes

4. Is the manuscript presented in an intelligible fashion and written in standard English?

Reviewer #1: Yes

5. Review Comments to the Author

Reviewer #1: In many ways this is an excellent and comprehensive report on the FHB population in South Africa. The authors conducted a large scale survey on the main wheat producing areas and expand the knowledge of Fusarium species and trichothecene chemotype composition in different ecological regions. The study has an outstanding level and should be published with some revisions.

1. The high frequency of FIESC in this study seems strange for me. Normally it is a very weak pathogenic species to wheat and widely distributed in soil, plant debris, surface of rice and wheat grains and so on. In my experience, for the wheat head samples with obvious symptoms, the isolation frequency would be lower than 1%, which can be ignored in the field FHB management practice. To my knowledge, there is also no report of such high ratio of FIESC as FHB pathogen before in other place worldwide. So the author should be careful about if they are real the causal agent of FHB or just surface contamination on the heads. Details of the isolation procedure should be examined. For example, for the heads with very mild symptom (i.e. without red mold), it would be hard to find the infected spikelet when they were dried several days after sampling, this maybe influence the result of the isolation. Anyway, I suggest the authors discuss the reasons of the unusual observation, if they are the pathogen, the high frequency of FIESC should considered in the field control and mycotoxin contamination.

2. P14L283-285, FCSC1, FCSC5 and FSSC5 were found. Did they determine by EF1a blast? These species were identified by phylogenetic analysis of several conserved genes, only ef1a sequence is identical?

3. The introduction and discussion are a bit long. I suggest to reduce some content not well fit the key points.

4. I noticed that the sample size of the places where F. pseudograminearum was predominant are small. The authors should mention it in the discussion, because small sample size is easy to cause bias.

5. Adding line numbers after page 21 will facilitate review.

6. PLOS authors have the option to publish the peer review history of their article (what does this mean?). If published, this will include your full peer review and any attached files.

Reviewer #1: **Yes: **Hao Zhang

---

## [Author Response · Author response to Decision Letter 0]

24 Dec 2021

Dear Reviewers

All comments have been addressed, as highlighted in the "Response to Reviewers". Please find attached the "Response to Reviewers", the "Revised Manuscript with Track Changes" and the "Manuscript", as requested.

I could not remove the original cover letter and manuscript from the submission (I take it that it must be standard practice for the journal), so those are also included in the Submission.

Regards,

Dr Gert J van Coller

---

## [Editor Report · Decision Letter 1]

12 Jan 2022

PONE-D-21-28634R1The distribution and type B trichothecene chemotype of *Fusarium* species associated with head blight of wheat in South AfricaPLOS ONE

Dear authors,

Thank you for submitting your manuscript to PLOS ONE. After careful consideration, we feel that it has merit but does not fully meet PLOS ONE’s publication criteria as it currently stands. Therefore, we invite you to submit a revised version of the manuscript that addresses the points raised during the review process.

We look forward to receiving your revised manuscript.

Kind regards,

Kris Audenaert, Ph.D.

Academic Editor

PLOS ONE

Journal Requirements:

Additional Editor Comments (if provided):

There is still one minor comment, in regard to table 3: the calculated incidence: "then calculated as follows: For FGSC: (30/35) x 100 = 85.71%, and for FIESC: (5/35) x

100 = 14.29%. This is also indicated in the subscript of Table 3." But my point is that when you have 100 sampels, you can never been more precise than 1%, so for example 1.567% is impossoble. This also accounts when you work with less samples, so everything lower than 1 shoudl definetely be removed....85.71% should be 85%...this is as precise as you can be when doing these types of calculation
---

## [Author Response · Author response to Decision Letter 1]

22 Feb 2022

Response to reviewers

Journal Requirements:

Comment: Please review your reference list to ensure that it is complete and correct. If you have cited papers that have been retracted, please include the rationale for doing so in the manuscript text, or remove these references and replace them with relevant current references. Any changes to the reference list should be mentioned in the rebuttal letter that accompanies your revised manuscript. If you need to cite a retracted article, indicate the article’s retracted status in the References list and also include a citation and full reference for the retraction notice.

Response: The reference list is complete and correct. All reference were checked on the PubMed database or the journal’s website to confirm that it has not been retracted. To the best of our knowledge, none of the references cited has been retracted. The personal communication with Richard Findlay (p. 29, line 535) has been replaced with two scientific articles (references 81 and 82), and the reference list has been updated accordingly. All Journal names have been edited to their correct abbreviations according to NCBI database, and all URLs have been removed.

Additional Editor Comments

Comment: There is still one minor comment, in regard to table 3: the calculated incidence: "then calculated as follows: For FGSC: (30/35) x 100 = 85.71%, and for FIESC: (5/35) x

100 = 14.29%. This is also indicated in the subscript of Table 3." But my point is that when you have 100 sampels, you can never been more precise than 1%, so for example 1.567% is impossoble. This also accounts when you work with less samples, so everything lower than 1 shoudl definetely be removed....85.71% should be 85%...this is as precise as you can be when doing these types of calculation.

Response: This has been addressed accordingly.

Reviewer’s Comments

Comment: While revising your submission, please upload your figure files to the Preflight Analysis and Conversion Engine (PACE) digital diagnostic tool, https://pacev2.apexcovantage.com/. PACE helps ensure that figures meet PLOS requirements. To use PACE, you must first register as a user. Registration is free. Then, login and navigate to the UPLOAD tab, where you will find detailed instructions on how to use the tool. If you encounter any issues or have any questions when using PACE, please email PLOS at figures@plos.org. Please note that Supporting Information files do not need this step.

Response: Figure files (Fig. 1 and Fig. 2) has been uploaded to the PACE digital diagnostic tool and has been archived.

---

## [Decision Letter · Decision Letter 2]

4 Jul 2022

PONE-D-21-28634R2The distribution and type B trichothecene chemotype of *Fusarium* species associated with head blight of wheat in South AfricaPLOS ONE

Dear Dr. Van Coller,

Thank you for submitting your manuscript to PLOS ONE. After careful consideration, we feel that it has merit but does not fully meet PLOS ONE’s publication criteria as it currently stands. Therefore, we invite you to submit a revised version of the manuscript that addresses the points raised during the review process.

We look forward to receiving your revised manuscript.

Kind regards,

Yuefeng Ruan, Ph.D

Academic Editor

PLOS ONE

Journal Requirements:

Reviewers' comments:

Reviewer's Responses to Questions

**Comments to the Author**

1. If the authors have adequately addressed your comments raised in a previous round of review and you feel that this manuscript is now acceptable for publication, you may indicate that here to bypass the “Comments to the Author” section, enter your conflict of interest statement in the “Confidential to Editor” section, and submit your "Accept" recommendation.

Reviewer #2: All comments have been addressed

Reviewer #3: All comments have been addressed

Reviewer #4: (No Response)

2. Is the manuscript technically sound, and do the data support the conclusions?

Reviewer #2: Yes

Reviewer #3: Yes

Reviewer #4: Yes

3. Has the statistical analysis been performed appropriately and rigorously? 

Reviewer #2: Yes

Reviewer #3: (No Response)

Reviewer #4: N/A

4. Have the authors made all data underlying the findings in their manuscript fully available?

Reviewer #2: Yes

Reviewer #3: Yes

Reviewer #4: (No Response)

5. Is the manuscript presented in an intelligible fashion and written in standard English?

Reviewer #2: Yes

Reviewer #3: Yes

Reviewer #4: Yes

6. Review Comments to the Author

Reviewer #2: Title and Abstract: a little bit misleading and unclear title. Since this survey was done in 14 years ago, it is better point out the time period in the title and abstract. Otherwise, audience will think this is a latest survey report.

Discussion: since this survey study is from 2008/09, almost 15 years ago. it is better to see if there is a latest survey report published recently and make a comparison.

Line 443: this is not a "current" study, but almost 15 years old

Reviewer #3: Dear Authors,

Your manuscript titled "The distribution and type B trichothecene chemotype of Fusarium species associated with head blight of wheat in South Africa" is a well written and structured manuscript that provides valuable information on the species associated with FHB infection. Unfortunately, the research and findings are outdated. Utilizing words such as "first report" is not accurate given that the sampling has been completed 13 and 14 years ago. The distribution pattern of species changes over the time while facing the changes of cropping system and practices, environment, the level of resistance of cultivars grown etc. I believe many changes could have happened in the last 14 years that could have changed the distribution pattern you are reporting in this study. Your manuscript still has a great value because it reports what species were predominant over a decade ago which could be a reference for newer studies. I strongly encourage you to address the findings as an overview research over the late 2000s.

Reviewer #4: This paper reports survey results conducted to determine the identity, distribution and type B trichothecene chemotype of Fusarium species associated with FHB of wheat in South Africa.

• General concern: The survey was in 2008 and 2009 which is about 13-14 year ago making it a historical data. So it doesn’t show recent developments although it still provides a good information if documented as a publication.

Abstract

o The finding shows chemotype diversity was limited (15-ADON=90.1%) and highly structure in terms of species difference. However, authors report for the first time six Fusarium species not recorded on wheat elsewhere which is a plus.

o Results were not indicated as related to production conditions – irrigated versus non-irrigated and province wise. One thing that should be addressed here is whether the species distribution under different farming conditions and provinces the same or different? Please describe it briefly.

Materials and methods

There is no mention if the locations overlapping across the two survey years. Please describe this as F. species differences between the years could be a factor of the sites of sample collection besides crop culture.

Results

o Line#273: What are the factors behind such large differences in the prevalence of FGSC between 2008 and 2009? This should be discussed in terms of sites of sample collection, weather condition, crop culture / rotation crops or species competition.

o Fig. 1. FGSC is common across all regions with one exception. In Western Cape and followed by Free State the proportion of F. pseudograminearum is exceptionally high. What makes this region so special. This needs discussing (seemingly due to bias in small sample size).

Discussion

o Lines#443-444: This sentence is confusing. Were these six Fusarium species not recorded on other crop species similar to wheat? If they reported on other plant species, they are not new species. They could be a new record on a wheat crop. Please reword the sentence.

Please check reference cited.

7. PLOS authors have the option to publish the peer review history of their article (what does this mean?). If published, this will include your full peer review and any attached files.

Reviewer #2: No

Reviewer #3: No

Reviewer #4: **Yes: **Firdissa Bokore

---

## [Author Response · Author response to Decision Letter 2]

17 Aug 2022

Response to reviewers

Journal Requirements:

Comment: Please review your reference list to ensure that it is complete and correct. If you have cited papers that have been retracted, please include the rationale for doing so in the manuscript text, or remove these references and replace them with relevant current references. Any changes to the reference list should be mentioned in the rebuttal letter that accompanies your revised manuscript. If you need to cite a retracted article, indicate the article’s retracted status in the References list and also include a citation and full reference for the retraction notice.

Response: The reference list is complete and correct. All references were checked on the PubMed database or the journal’s website to confirm that it has not been retracted. To the best of our knowledge, none of the references cited has been retracted.

Reviewer’s comments:

6. Review Comments to the Author

Reviewer #2:

• Title and Abstract: a little bit misleading and unclear title. Since this survey was done in 14 years ago, it is better point out the time period in the title and abstract. Otherwise, audience will think this is a latest survey report.

Response: The title has been changed to reflect the time period in which the survey was conducted. This time period is also indicated in the Abstract. The Abstract has been reduced to 300 words, in accordance to Journal requirements.

• Discussion: since this survey study is from 2008/09, almost 15 years ago. it is better to see if there is a latest survey report published recently and make a comparison.

Response: To the best of our knowledge, from the time this survey was conducted, no newer surveys were done or reported in South Africa. Our survey is, therefore, the latest and most complete report on Fusarium species associated with FHB of wheat in South Africa to date. All previous surveys done in South Africa, and results obtained, are mentioned in the Introduction and thoroughly discussed in relation to our findings in the Discussion. In the most recently published survey (Minnaar-Ontong et al., 2017), no mention is made of when their study was conducted, while the isolates they obtained was identified using morphological characteristics only. Minnaar-Ontong et al. (2017) also obtained samples from fewer locations and production regions than our study. Every study to date conducted under South African conditions is comprehensively referred to in the Discussion.

• Line 443: this is not a "current" study, but almost 15 years old

Response: This has been addressed (line 449-450 in the Revised Manuscript with Track Changes – “Simple Markup” in the “Review” Tab).

Reviewer #3:

• Your manuscript titled "The distribution and type B trichothecene chemotype of Fusarium species associated with head blight of wheat in South Africa" is a well written and structured manuscript that provides valuable information on the species associated with FHB infection. Unfortunately, the research and findings are outdated. Utilizing words such as "first report" is not accurate given that the sampling has been completed 13 and 14 years ago. The distribution pattern of species changes over the time while facing the changes of cropping system and practices, environment, the level of resistance of cultivars grown etc. I believe many changes could have happened in the last 14 years that could have changed the distribution pattern you are reporting in this study. Your manuscript still has a great value because it reports what species were predominant over a decade ago which could be a reference for newer studies. I strongly encourage you to address the findings as an overview research over the late 2000s.

Response: Just prior to first submitting this manuscript to PLOS ONE on 2021/09/03, the generated EF1-α sequences were compared to existing sequences in the databases mentioned in the Materials and Methods, in collaboration with our co-author, Dr. Todd Ward (Director – ARS Culture Collection (NRRL)) from the USDA. Results were then compared to existing literature, to ensure that the first reports of Fusarium species mentioned in the manuscript are valid. That is how, for example, the first report of Fusarium transvaalense from wheat was provided, even though this species was only described in 2018 (reference # 70: Sandoval-Denis et al., 2018). This is also how the potentially novel Fusarium spp. from the FCSC, FFSC, FOSC, FSAMSC and FTSC was identified.

The Fusarium species associated with FHB in South Africa could have changed in response to changes of cropping system and practices, environment, the level of resistance of cultivars grown etc. Therefore, and due to the fact that our study was conducted from 2008 – 2009, care is taken to present our study as originating from the late 2000s, while emphasising the need for new studies, since our study is still the latest and most complete survey done in South Africa to date. (see Abstract, Introduction (lines 137-140), Materials and Methods (lines 160-162 and 176-180) and Discussion (lines 438-439, 473-475, 480-490, 494-500, 501-503, 574-589, 590-598) in the Revised Manuscript with Track Changes – “Simple Markup” in the “Review” Tab).

Reviewer #4:

This paper reports survey results conducted to determine the identity, distribution and type B trichothecene chemotype of Fusarium species associated with FHB of wheat in South Africa.

• General concern: The survey was in 2008 and 2009 which is about 13-14 year ago making it a historical data. So it doesn’t show recent developments although it still provides a good information if documented as a publication.

Response: No response needed.

Abstract:

• The finding shows chemotype diversity was limited (15-ADON=90.1%) and highly structure in terms of species difference. However, authors report for the first time six Fusarium species not recorded on wheat elsewhere which is a plus.

Response: No response needed.

• Results were not indicated as related to production conditions – irrigated versus non-irrigated and province wise. One thing that should be addressed here is whether the species distribution under different farming conditions and provinces the same or different? Please describe it briefly.

Response: This has been addressed in the Abstract (lines 33-35 in the Revised Manuscript with Track Changes – “Simple Markup” in the “Review” Tab). The Abstract has also been reduced to 300 words, in accordance to Journal requirements.

Materials and Methods:

• There is no mention if the locations overlapping across the two survey years. Please describe this as F. species differences between the years could be a factor of the sites of sample collection besides crop culture.

Response: This has been addressed (lines 162-165 in the Revised Manuscript with Track Changes – “Simple Markup” in the “Review” Tab).

Results:

• Line#273: What are the factors behind such large differences in the prevalence of FGSC between 2008 and 2009? This should be discussed in terms of sites of sample collection, weather condition, crop culture / rotation crops or species competition.

Response: This has been addressed in the Discussion (lines 574-587 in the Revised Manuscript with Track Changes – “Simple Markup” in the “Review” Tab):

“FGSC and FIESC isolates were more abundant in the Free State in 2008 compared to 2009. This also happened for the FGSC in the Northern Cape. In KZN, the occurrence of the FGSC remained almost unchanged during the two years. The presence of all members of FCSC, F. temperatum and F. verticillioides, F. oxysporum (FOSC), F. armeniacum, F. culmorum, F. lunulosporum, F. transvaalense, FSSC 5, F. acuminatum, F. avenaceum, and an unknown Fusarium sp. within FTSC in 2008 but not 2009, and F. subglutinans, F. brachygibbosum, and unknown Fusarium spp. within FFSC and FSAMSC in 2009 but not 2008, may have been coincidental since the incidence of all of these species was very low. The high incidence of Alternaria species obtained in all production regions in 2009 might have contributed to lower Fusarium levels in the 2009 production season, while differences in climate, cropping history and agronomic practices of the collection sites in the two years might have also contributed to the discrepancy in Fusarium species composition between years [40, 41, 43]. Variation in the timing and the amount of water provided through irrigation, especially near anthesis, could also have influenced the resultant disease intensity and associated Fusarium species between years [87].”

• Fig. 1. FGSC is common across all regions with one exception. In Western Cape and followed by Free State the proportion of F. pseudograminearum is exceptionally high. What makes this region so special. This needs discussing (seemingly due to bias in small sample size).

Response: This has been addressed in the Discussion (lines 501-512 in the Revised Manuscript with Track Changes – “Simple Markup” in the “Review” Tab):

“An interesting observation was the dominance of F. pseudograminearum as an FHB pathogen at one locality in the eastern Free State and the one locality in the Western Cape. The dominance of this species at Bethlehem in the eastern Free State may be due to it being introduced earlier via infected seed [35, 85], especially since F. pseudograminearum was not found at any other locality in this region. Although the sample size in the Western Cape was quite small (one locality with 40 wheat heads sampled), a study conducted more recently from three localities in the Western Cape revealed that F. pseudograminearum was the dominant species obtained from wheat heads exhibiting FHB symptoms at all three localities, constituting more than 80% of ~300 isolates obtained (unpublished data). Fusarium pseudograminearum is best known as the cause of Fusarium crown rot (FCR) of wheat [73, 74]. Its dominance in the Western Cape can be ascribed to the prevalence of FCR in this region as well as the use of minimum / no till practices, which results in a build-up of inoculum levels in stubble [74].”

The more recent study, in which ~300 F. pseudograminearum isolates was found from FHB at three localities in the Western Cape (unpublished data), forms part of a population genetics study currently being conducted within our research group, which will be published in future.

• Lines#443-444: This sentence is confusing. Were these six Fusarium species not recorded on other crop species similar to wheat? If they reported on other plant species, they are not new species. They could be a new record on a wheat crop. Please reword the sentence.

Response: This has been addressed (line 449-450 in the Revised Manuscript with Track Changes – “Simple Markup” in the “Review” Tab).

• Please check references cited.

Response: The references cited in the text and reference list are complete.

---

## [Decision Letter · Decision Letter 3]

12 Sep 2022

The distribution and type B trichothecene chemotype of *Fusarium* species associated with head blight of wheat in South Africa during 2008 and 2009

PONE-D-21-28634R3

Dear Dr. Van Coller,

We’re pleased to inform you that your manuscript has been judged scientifically suitable for publication and will be formally accepted for publication once it meets all outstanding technical requirements.

Kind regards,

Yuefeng Ruan, Ph.D

Academic Editor

PLOS ONE

Additional Editor Comments (optional):

Dear Authors:

Thank you for addressing reviewers’ comments and revising the manuscript.

Thank you for submitting to PLOS ONE again.

Reviewers' comments:

Reviewer's Responses to Questions

**Comments to the Author**

1. If the authors have adequately addressed your comments raised in a previous round of review and you feel that this manuscript is now acceptable for publication, you may indicate that here to bypass the “Comments to the Author” section, enter your conflict of interest statement in the “Confidential to Editor” section, and submit your "Accept" recommendation.

Reviewer #4: All comments have been addressed

2. Is the manuscript technically sound, and do the data support the conclusions?

Reviewer #4: Yes

3. Has the statistical analysis been performed appropriately and rigorously? 

Reviewer #4: Yes

4. Have the authors made all data underlying the findings in their manuscript fully available?

Reviewer #4: Yes

5. Is the manuscript presented in an intelligible fashion and written in standard English?

Reviewer #4: Yes

6. Review Comments to the Author

Reviewer #4: The authors responded to my comments and made improvements to the previous version. I have no further comments to make.

7. PLOS authors have the option to publish the peer review history of their article (what does this mean?). If published, this will include your full peer review and any attached files.

Reviewer #4: **Yes: **Firdissa Bokore

---

## [Editor Report · Acceptance letter]

14 Sep 2022

PONE-D-21-28634R3 

The distribution and type B trichothecene chemotype of *Fusarium* species associated with head blight of wheat in South Africa during 2008 and 2009 

Dear Dr. Van Coller:

I'm pleased to inform you that your manuscript has been deemed suitable for publication in PLOS ONE. Congratulations! Your manuscript is now with our production department. 

Kind regards, 

on behalf of

Dr. Yuefeng Ruan 

Academic Editor

PLOS ONE